# MixUp as Directional Adversarial Training

## Abstract

MixUp is a data augmentation scheme in which pairs of training samples and their corresponding labels are mixed using linear coefficients. Without label mixing, MixUp becomes a more conventional scheme: input samples are moved but their original labels are retained. Because samples are preferentially moved in the direction of other classes we refer to this method as directional adversarial training, or DAT. We show that under two mild conditions, MixUp asymptotically convergences to a subset of DAT. We define untied MixUp (UMixUp), a superset of MixUp wherein training labels are mixed with different linear coefficients to those of their corresponding samples. We show that under the same mild conditions, untied MixUp converges to the entire class of DAT schemes. Motivated by the understanding that UMixUp is both a generalization of MixUp and a form of adversarial training, we experiment with different datasets and loss functions to show that UMixUp provides improved performance over MixUp. In short, we present a novel interpretation of MixUp as belonging to a class highly analogous to adversarial training, and on this basis we introduce a simple generalization which outperforms MixUp.

## 1 Introduction

Deep learning applications often require complex networks with a large number of parameters (He et al., 2016; Zagoruyko & Komodakis, 2016; Devlin et al., 2018). Although neural networks perform so well that their ability to generalize is an area of study in itself (Zhang et al., 2017a; Arpit et al., 2017), their high complexity nevertheless causes them to overfit their training data (Kukacka et al., 2017). For this reason, effective regularization techniques are in high demand.

There are two flavors of regularization: complexity curtailing and data augmentation [1]. Complexity curtailing methods constrain models to learning in a subset of parameter space which has a higher probability of generalizing well. Notable examples are weight decay (Krogh & Hertz, 1991) and dropout (Srivastava et al., 2014).

Data augmentation methods add transformed versions of training samples to the original training set. Conventionally, transformed samples retain their original label, so that models effectively see a larger set of data-label training pairs. Commonly applied transformations in image applications include flips, crops and rotations.

A recently devised family of augmentation schemes called adversarial training has attracted active research interest (Szegedy et al., 2013; Goodfellow et al., 2014; Miyato et al., 2016; Athalye et al., 2018; Shaham et al., 2018; He et al., 2018). Adversarial training seeks to reduce a model's propensity to misclassify minimally perturbed training samples, or adversarials. While attack algorithms used for testing model robustness may search for adversarials in unbounded regions of input space, adversarial training schemes generally focus on perturbing training samples within a bounded region, while retaining the sample's original label (Goodfellow et al., 2015; Shaham et al., 2018).

Another recently proposed data augmentation scheme is MixUp (Zhang et al., 2017b), in which new samples are generated by mixing pairs of training samples using linear coefficients. Despite its well established generalization performance (Zhang et al., 2017b; Guo et al., 2018; Verma et al., 2018), the working mechanism of MixUp is not well understood. Guo et al. (2018) suggest viewing MixUp as imposing local linearity on the model using points outside of the data manifold. While this

---

[1] Some authors describe these flavors as "data independent" and "data-dependent" (Guo et al., 2018).

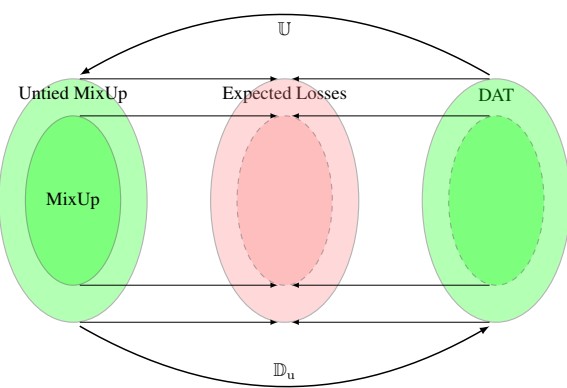

Figure 1: The relationship between MixUp, DAT and Untied MixUp.

perspective is insightful, we do not believe it paints a full picture of how MixUp operates. A recent study (Lamb et al., 2019) provides empirical evidence that MixUp improves adversarial robustness, but does not present MixUp as a form of adversarial training.

We build a framework to understand MixUp in a broader context: we argue that adversarial training is a central working principle of MixUp. To support this contention, we connect MixUp to a MixUp-like scheme which does not perform label mixing, and we relate this scheme to adversarial training.

Without label mixing, MixUp becomes a conventional augmentation scheme: input samples are moved, but their original labels are retained. Because samples are moved in the direction of other samples – which are typically clustered in input space – we describe this method as 'directional'. Because this method primarily moves training samples in the direction of adversarial classes, this method is analogous to adversarial training. We thus refer to MixUp without label mixing as directional adversarial training (DAT). We show that MixUp converges to a subset of DAT under mild conditions, and we thereby argue that adversarial training is a working principle of MixUp.

Inspired by this new understanding of MixUp as a form of adversarial training, and upon realizing that MixUp is (asymptotically) a subset of DAT, we introduce Untied MixUp (UMixUp), a simple enhancement of MixUp which converges to the entire family of DAT schemes, as depicted in Figure 1. Untied Mixup mixes data-label training pairs in a similar way to MixUp, with the distinction that the label mixing ratio is an arbitrary function of the sample mixing ratio. We perform experiments to show that UMixUp's classification performance improves upon MixUp.

In short, this research is motivated by a curiosity to better understand the working of MixUp. In-so-doing we aim to:

1. **Establish DAT as analogous to adversarial training**. This is discussed in section 4.

2. **Establish UMixUp as a superset of MixUp**, and as converging to the entire family of DAT schemes. In-so-doing, a) establish MixUp's convergence to a subset of DAT, and thereby that it operates analogously to adversarial training; and b) establish UMixUp as a broader class of MixUp-like schemes that operate analogously to adversarial training. This is discussed in 5.

3. **Establish empirically that UMixUp's classification performance improves upon MixUp.** This is discussed in section 6.

Finally we note that this paper has another contribution. Conventionally, MixUp is only applicable to baseline models that use cross entropy loss. All analytical results we develop in this paper are applicable to a wider family of models using any loss function which we term target-linear. We define target-linearity and experiment with a new loss function called negative cosine-loss to show its potential.

Essential proofs of theoretical results are given in the Appendix.

## 2 PRELIMINARIES

Column vectors are denoted by bold letters such as $\mathbf{m}$, and sets are denoted by calligraphic uppercase letters such as $\mathcal{M}$. The component of a vector is denoted by a bracketed index. For example, $\mathbf{m}[i]$ denotes the $i$th component of $\mathbf{m}$.

Regular (non-calligraphic) capitalized letters such as $\mathbf{X}$ will denote random variables, and their lowercase counterparts, e.g., $\mathbf{x}$, will denote realizations of a random variable. Any sequence, $(a_1, a_2, \ldots, a_n)$ will be denoted by $a_1^n$. Likewise $(A_1, A_2, \ldots, A_n)$ will be denoted by $A_1^n$, and a sequence of sample pairs $((\mathbf{x}_1, \mathbf{x}_1'), (\mathbf{x}_2, \mathbf{x}_2'), \ldots, (\mathbf{x}_n, \mathbf{x}_n'))$ denoted by $(\mathbf{x}, \mathbf{x}')_1^n$.

For any value $a \in [0, 1]$, we will use $\overline{a}$ as a short notation for $1 - a$.

**Classification Setting** Consider a standard classification problem, in which one wishes to learn a classifier that predicts the *class label* for a *sample*.

Formally, let $\mathcal{X}$ be a vector space in which the samples of interest live and let $\mathcal{Y}$ be the set of all possible labels associated with these samples. The set of training samples will be denoted by $\mathcal{D}$, a subset of $\mathcal{X}$. We will use $t(\mathbf{x})$ to denote the true label of $\mathbf{x}$. Let $F$ be a neural network function, parameterized by $\theta$, which maps $\mathcal{X}$ to another vector space $\mathcal{Z}$. Let $\varphi : \mathcal{Y} \to \mathcal{Z}$ be a function that maps a label in $\mathcal{Y}$ to an element in $\mathcal{Z}$ such that for any $y, y' \in \mathcal{Y}$, if $y \neq y'$, then $\varphi(y) \neq \varphi(y')$. In the space $\mathcal{Z}$, we refer to $F(\mathbf{x})$ as the *model's prediction*. With slight abuse of language, we will occasionally refer to both $t(\mathbf{x})$ and $\varphi(t(\mathbf{x}))$ as the "label" of $\mathbf{x}$. Let $\ell : \mathcal{Z} \times \mathcal{Z} \to \mathbb{R}$ be a *loss function*, using which one defines an *overall loss function* as

$$\mathcal{L} := \frac{1}{|\mathcal{D}|} \sum_{x \in \mathcal{D}} \ell\left(F(\mathbf{x}), \varphi(t(\mathbf{x}))\right) \tag{1}$$

Here we have taken the notational convention that the first argument of $\ell$ represents the model's prediction and the second represents the target label. In this setting, the learning problem is formulated as minimizing $\mathcal{L}$ with respect to its characterizing parameters $\theta$.

**Target-Linear Loss Functions** We say that a loss function $\ell(\mathbf{z}, \mathbf{z}')$ is *target-linear* if for any scalars $\alpha$ and $\beta$,

$$\ell(\mathbf{z}, \alpha\mathbf{z}_1 + \beta\mathbf{z}_2) = \alpha\ell(\mathbf{z}, \mathbf{z}_1) + \beta\ell(\mathbf{z}, \mathbf{z}_2)$$

Target-linear loss functions arise naturally in many settings, for which we now provide two examples. For convenience, we define the vectors $\mathbf{v} = F(\mathbf{x})$ and $\mathbf{y} = \varphi(t(\mathbf{x}))$.

Cross-Entropy Loss The conventional cross-entropy loss function, written in our notation, is defined as:

$$\ell_{\text{CE}}\left(F(\mathbf{x}), \varphi(t(\mathbf{x}))\right) = \ell_{\text{CE}}(\mathbf{v}, \mathbf{y}) := \sum_{i=1}^{dim(\mathcal{Z})} \mathbf{y}[i] \log \mathbf{v}[i]$$

where $\mathbf{v}$ and $\mathbf{y}$ are constrained to being probability vectors. We note that in conventional applications, $dim(\mathcal{Z}) = |\mathcal{Y}|$, and the target label $\mathbf{v}$ is a one-hot vector where $\mathbf{y}[i] = 1$ if $i = t(\mathbf{x})$ and $\mathbf{y}[i] = 0$ otherwise. Constraining $\mathbf{v}$ to being a probability vector is achieved using a softmax output layer.

Negative-Cosine Loss The "negative-cosine loss", usually used in its negated version, i.e., as the cosine similarity, can be defined as follows.

$$\ell_{\text{NC}}\left(F(\mathbf{x}), \varphi(t(\mathbf{x}))\right) = \ell_{\text{NC}}(\mathbf{v}, \mathbf{y}) := -\mathbf{v}^{\text{T}}\mathbf{y}$$

where $\mathbf{v}$ and $\mathbf{y}$ are constrained to being unit-length vectors. For $\mathbf{v}$ this can be achieved by simple division at the output layer, and for $\mathbf{y}$ by limiting the range of $\varphi$ to an orthonormal basis (making it a conventional label embedding function).

It is clear that the cross-entropy loss $\ell_{\text{CE}}$ and the negative-cosine loss $\ell_{\text{NC}}$ are both target-linear, directly following from the definition of target-linearity.

**Assumptions** The theoretical development of this paper relies on two fundamental assumptions, which we call "axioms".

**Axiom 1** *(Target linearity) The loss function $\ell$ used for the classification setting is target-linear.*

That is, the study of MixUp in this paper is in fact goes beyond the standard MixUp, which uses the cross-entropy loss.

Much of the development in this paper concerns drawing sample pairs $(\mathbf{x}, \mathbf{x}')$ from $\mathcal{D} \times \mathcal{D}$. Suppose that $(\mathbf{x}, \mathbf{x}')_1^n$ is a length-$n$ sequence of sample pairs drawn from $\mathcal{D} \times \mathcal{D}$. A sequence $(\mathbf{x}, \mathbf{x}')_1^n$ is said to be symmetric if for every $(\mathbf{a}, \mathbf{b}) \in \mathcal{D} \times \mathcal{D}$, the number of occurrences of $(\mathbf{a}, \mathbf{b})$ in the sequence is equal to that of $(\mathbf{b}, \mathbf{a})$. A distribution $Q$ on $\mathcal{D} \times \mathcal{D}$ will be called *exchangeable*, or *symmetric*, if for any $(\mathbf{x}, \mathbf{x}') \in \mathcal{D} \times \mathcal{D}$, $Q((\mathbf{x}, \mathbf{x}')) = Q((\mathbf{x}', \mathbf{x}))$.

**Axiom 2** *(Symmetric pair-sampling distribution) Whenever a sample pair $(\mathbf{x}, \mathbf{x}')$ is drawn from a distribution $Q$, $Q$ is assumed to be symmetric.*

In the standard MixUp, two samples are drawn independently from $\mathcal{D}$ to form a pair, making this condition satisfied.

## 3    MIXUP, DAT, UNTIED MIXUP

### 3.1    INFORMAL SUMMARY

We first provide a summary of each scheme for the reader's convenience. We then describe each scheme more systematically. For concision of equations to follow, we define

$$\mathbf{y} = \varphi(t(\mathbf{x})) \qquad \text{and} \qquad \mathbf{y}' = \varphi(t(\mathbf{x}'))$$

MixUp is a data augmentation scheme in which samples are linearly combined using some mixing ratio $\lambda \in [0, 1]$:

$$\mathbf{x}_g = \lambda \mathbf{x} + (1 - \lambda)\mathbf{x}' \tag{2}$$

where $\lambda \sim P^{\text{Mix}}$. A target label is generated using the same mixing ratio $\lambda$:

$$\mathbf{y}_g = \lambda \mathbf{y} + (1 - \lambda)\mathbf{y}' \tag{MixUp}$$

DAT and UMixUp use the same method (2) for generating samples, but use different $\lambda$ distributions ($P^{\text{DAT}}$ and $P^{\text{uMix}}$ respectively). DAT and UMixUp also differ from MixUp in their target labels. DAT retains the sample's original label:

$$\mathbf{y}_g = \mathbf{y} \tag{DAT}$$

whereas UMixUp's label mixing ratio is a function of $\lambda$:

$$\mathbf{y}_g = \gamma(\lambda)\mathbf{y} + (1 - \gamma(\lambda))\mathbf{y}' \tag{UMixUp}$$

In Untied MixUp, the label mixing ratio is "untied" from the sample mixing ratio, and can be any $\gamma(\lambda)$. We will refer to $\gamma$ as the *weighting function*. An Untied MixUp scheme is specified both by the its mixing policy $P^{\text{uMix}}$ and a weighting function $\gamma$.

### 3.2    FORMAL DEFINITIONS

To draw comparisons between MixUp, DAT, and Untied MixUp schemes, we establish a framework for characterizing their optimization problems. To that end, we define each model's loss function $\ell^m$ in terms of its baseline target-linear loss function $\ell_b$, where the superscript $m$ is replaced with a model identifier (i.e. $\ell^{\text{Mix}}$, $\ell^{\text{DAT}}$, $\ell^{\text{uMix}}$). Each model's overall loss function, $\mathcal{L}^m$ is defined in terms of its loss function $\ell^m$ as per equation 1 (where equation 1's $\ell$ is $\ell^m$). We denote the expected value of each scheme's overall loss, $\mathcal{L}_E^m$, with respect to its mixing ratio $\Lambda$.

Let $n$ be a positive integer. In every scheme, a sequence $(\mathbf{x}, \mathbf{x}')_1^n := ((\mathbf{x}_1, \mathbf{x}_1'), (\mathbf{x}_2, \mathbf{x}_2'), \ldots, (\mathbf{x}_n, \mathbf{x}_n'))$ of sample pairs are drawn i.i.d. from $Q$, and a sequence $\lambda_1^n := (\lambda_1, \lambda_2, \ldots, \lambda_n)$ of values are drawn i.i.d. from $P^m$, where $P^m$ is a distribution over $[0, 1]$ unique to each model.

**MixUp** For any $\mathbf{x}, \mathbf{x}' \in \mathcal{D}$ and any $\lambda \in [0, 1]$, denote

$$\ell^{\text{Mix}}(\mathbf{x}, \mathbf{x}', \lambda) := \ell_b \left( F(\lambda \mathbf{x} + \overline{\lambda} \mathbf{x}'), \lambda \mathbf{y} + \overline{\lambda} \mathbf{y}' \right)$$

Let $P^m$ be $P^{\text{Mix}}$; in other words a sequence $\lambda_1^n := (\lambda_1, \lambda_2, \ldots, \lambda_K)$ of values are drawn i.i.d. from $P^{\text{Mix}}$. We denote the overall loss $\mathcal{L}^{\text{Mix}}((\mathbf{x}, \mathbf{x}')_1^n, \lambda_1^n)$ and the expected overall loss $\mathcal{L}_E^{\text{Mix}}((\mathbf{x}, \mathbf{x}')_1^n)$:

$$\mathcal{L}^{\text{Mix}}((\mathbf{x}, \mathbf{x}')_1^n, \lambda_1^n) := \frac{1}{n} \sum_{k=1}^{n} \ell^{\text{Mix}}(\mathbf{x}_k, \mathbf{x}'_k, \lambda_k) \tag{3}$$

$$\mathcal{L}_E^{\text{Mix}}((\mathbf{x}, \mathbf{x}')_1^n) := \mathbb{E}_{\lambda_1^n \overset{\text{iid}}{\sim} P^{\text{Mix}}} \mathcal{L}^{\text{Mix}}((\mathbf{x}, \mathbf{x}')_1^n, \lambda_1^n)$$

In MixUp, we refer to $P^{\text{Mix}}$ as the *mixing policy*.

**Directional Adversarial Training (DAT)** For any $\mathbf{x}, \mathbf{x}' \in \mathcal{D}$ and any $\lambda \in [0, 1]$, we denote

$$\ell^{\text{DAT}}(\mathbf{x}, \mathbf{x}', \lambda) := \ell_b \left( F(\lambda \mathbf{x} + \overline{\lambda} \mathbf{x}'), \mathbf{y} \right)$$

Let $P^m$ be $P^{\text{DAT}}$, such that members of $\lambda_1^n$ are drawn i.i.d. from $P^{\text{DAT}}$. We denote the overall loss $\mathcal{L}^{\text{DAT}}((\mathbf{x}, \mathbf{x}')_1^n, \lambda_1^n\})$ and the expected overall loss $\mathcal{L}_E^{\text{DAT}}((\mathbf{x}, \mathbf{x}')_1^n)$:

$$\mathcal{L}^{\text{DAT}}((\mathbf{x}, \mathbf{x}')_1^n, \lambda_1^n\}) := \frac{1}{n} \sum_{k=1}^{n} \ell^{\text{DAT}}(\mathbf{x}_k, \mathbf{x}'_k, \lambda_k) \tag{4}$$

$$\mathcal{L}_E^{\text{DAT}}((\mathbf{x}, \mathbf{x}')_1^n) := \mathbb{E}_{\lambda_1^n \overset{\text{iid}}{\sim} P^{\text{DAT}}} \mathcal{L}^{\text{DAT}}((\mathbf{x}, \mathbf{x}')_1^n, \lambda_1^n)$$

In DAT, we refer to $P^{\text{DAT}}$ as the *adversarial policy*.

**Untied MixUp (UMixUp)**

Let $\gamma$ be a function mapping $[0, 1]$ to $[0, 1]$. For any $\mathbf{x}, \mathbf{x}' \in \mathcal{D}$ and any $\lambda \in [0, 1]$, we denote

$$\ell^{\text{uMix}}(\mathbf{x}, \mathbf{x}', \lambda, \gamma) := \ell_b(F(\lambda \mathbf{x} + \overline{\lambda} \mathbf{x}'), \gamma(\lambda)\mathbf{y} + \overline{\gamma(\lambda)}\mathbf{y}')$$

Let $P^m$ be $P^{\text{uMix}}$, and denote the overall and expected overall loss functions $\mathcal{L}^{\text{uMix}}((\mathbf{x}, \mathbf{x}')_1^n, \lambda_1^n, \gamma)$ and $\mathcal{L}_E^{\text{uMix}}((\mathbf{x}, \mathbf{x}')_1^n, \gamma)$ respectively:

$$\mathcal{L}^{\text{uMix}}((\mathbf{x}, \mathbf{x}')_1^n, \lambda_1^n, \gamma) := \frac{1}{n} \sum_{k=1}^{n} \ell^{\text{uMix}}(\mathbf{x}, \mathbf{x}', \lambda, \gamma)$$

$$\mathcal{L}_E^{\text{uMix}}((\mathbf{x}, \mathbf{x}')_1^n, \gamma) := \mathbb{E}_{\lambda_1^n \overset{\text{iid}}{\sim} P^{\text{uMix}}} \mathcal{L}^{\text{uMix}}((\mathbf{x}, \mathbf{x}')_1^n, \lambda_1^n, \gamma)$$

At this end, it is apparent that MixUp is a special case of Untied MixUp, where the function $\gamma(\lambda)$ takes the simple form $\gamma(\lambda) = \lambda$.

## 4 DAT AS ANALOGOUS TO ADVERSARIAL TRAINING

The main theoretical result of this paper is the relationship established between DAT and UMixUp, and by extension MixUp. Both MixUp and UMixUp will be shown to converge to DAT as the number of mixed sample pairs, $n$, tends to infinity. Prior to developing these results, we provide insight into DAT, in terms of its similarity to adversarial training and its regularization mechanisms.

Conventional adversarial training schemes augment the original training dataset by searching for approximations of true adversarials within bounded regions around each training sample. For a training sample $\mathbf{x}$, a bounded region $U$ known as an $L_p$ ball is defined as $U = \{\mathbf{x} + \boldsymbol{\eta} : ||\boldsymbol{\eta}||_p < \epsilon\}$. Over this region, the loss function with respect to the true label of $\mathbf{x}$ is maximized. A typical loss function for an adversarial scheme is

$$\ell(F(\mathbf{x}), \mathbf{y}) = \max_{\tilde{\mathbf{x}} \in U} \ell_b(F(\tilde{\mathbf{x}}), \mathbf{y})$$

where $\ell_b$ is the baseline loss function. Simply put, baseline training serves to learn correct classification over the training data, whereas adversarial training moves the classification boundary to improve generalization.

DAT, on the other hand, combines intra-class mixing (mixing two samples of the same class) and inter-class mixing (mixing samples of different classes). Intra-class mixing serves to smooth classification boundaries of inner-class regions, while inter-class mixing perturbs training samples in the direction of adversarial classes, which improves generalization. Inter-class mixing dwarves intra-class mixing by volume of generated samples seen by the learning model in most many-class learning problems (by a 9-1 ratio in balanced 10-class problems for instance). DAT, which primarily consists of inter-class mixing, can therefore be seen as analogous to adversarial training.

The key distinction between conventional adversarial training and inter-class mixing is that MixUp movement is determined probabilistically within a bounded region, while adversarial movement is deterministic.

Figure 2 illustrates the connection between standard adversarial training and DAT. Consider the problem of classifying the blue points and the black points in Figure 2a), where the dashed curve is a ground-truth classifier and the black curve indicates the classification boundary of $F(\mathbf{x})$, which overfits the training data. In adversarial training, a training sample $\mathbf{x}$ is moved to a location within an $L_p$-ball around $\mathbf{x}$ while keeping its label to further train the model; the location, denoted by $\widehat{\mathbf{x}}_1$ in Figure 2b), is chosen to maximize training loss.

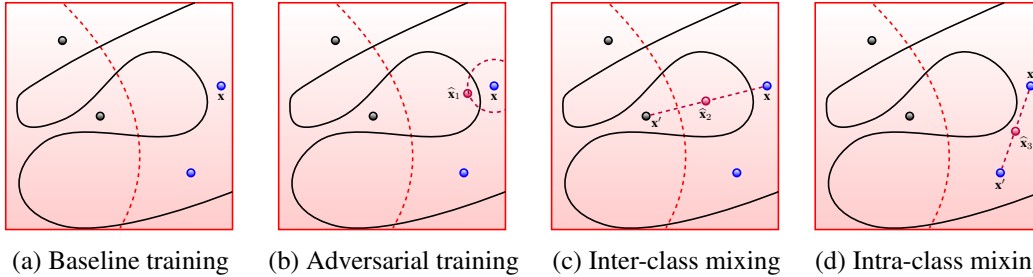

| (a) Baseline training | (b) Adversarial training | (c) Inter-class mixing | (d) Intra-class mixing |

Figure 2: DAT as a form of adversarial training

In DAT, a second sample $\mathbf{x}'$ governs the direction in which $\mathbf{x}$ is perturbed. If $\mathbf{x}'$ is chosen from a different class as shown in Figure 2c), then the generated sample $\widehat{\mathbf{x}}_2$ is used to further train the model. If $\mathbf{x}'$ is chosen from the same class as shown in Figure 2d), then the sample $\widehat{\mathbf{x}}_3$ is used in further training. Note that the inter-class mixed sample $\widehat{\mathbf{x}}_2$ pushes the model's classification boundary closer to the ground-truth classifier, thus connecting adversarial training and DAT. The intra-class sample $\widehat{\mathbf{x}}_3$, on the other hand, mainly helps to smooth inner parts of the class region. The latter behaviour is an additional feature of DAT and MixUp, which distinguishes these schemes from adversarial training.

## 5 UNTIED MIXUP AS ASYMPTOTICALLY EQUIVALENT TO DAT

We now show that Untied MixUp and DAT are equivalent when $n$ tends to infinity. A consequence of this equivalence is that it infuses both MixUp and UMixUp with the intuition of adversarial training. To that end, we relate the Untied MixUp loss function, $\ell^{\mathrm{uMix}}$, with the DAT loss function, $\ell^{\mathrm{DAT}}$.

**Lemma 1** *For any* $(\mathbf{x}, \mathbf{x}') \in \mathcal{D} \times \mathcal{D}$ *and any* $\lambda \in [0, 1]$,

$$\ell^{\mathrm{uMix}}(\mathbf{x}, \mathbf{x}', \lambda, \gamma) = \gamma(\lambda)\ell^{\mathrm{DAT}}(\mathbf{x}, \mathbf{x}', \lambda) + \overline{\gamma(\lambda)}\ell^{\mathrm{DAT}}(\mathbf{x}', \mathbf{x}, \overline{\lambda})$$

This result follows directly from the target-linearity of the loss function.

The next two lemmas show that as $n$ tends to infinity, the overall loss of both DAT and UMixUp converge in probability to their respective overall expected losses.

**Lemma 2** *As* $n$ *increases,* $\mathcal{L}^{\mathrm{DAT}}\left((\mathbf{x}, \mathbf{x}')_1^n, \Lambda_1^n\right)$ *converges to* $\mathcal{L}_E^{\mathrm{DAT}}\left((\mathbf{x}, \mathbf{x}')_1^n\right)$ *in probability.*

**Lemma 3** *As $n$ increases, $\mathcal{L}^{\mathrm{uMix}}\left((\mathbf{x},\mathbf{x}')_1^n,\Lambda_1^n,\gamma\right)$ converges to $\mathcal{L}_E^{\mathrm{uMix}}\left((\mathbf{x},\mathbf{x}')_1^n,\gamma\right)$ in probability.*

These two lemmas have similar proofs, thus only the proof of Lemma 2 is given in section A.1.

Next we show that as $n$ tends to infinity, UMixUp converges in probability to a subset of DAT, and DAT converges in probability to a subset of UMixUp. In other words, we show that as $n$ increases, UMixUp converges to being equivalent to the entire class of DAT schemes.

For that purpose, let $\mathcal{F}$ denote the space of all functions mapping $[0,1]$ to $[0,1]$. Each configuration in $\mathcal{P}\times\mathcal{F}$ defines an Untied MixUp scheme.

We now define $\mathbb{U}$, which maps a DAT scheme to an Untied MixUp scheme. Specifically $\mathbb{U}$ is a map from $\mathcal{P}$ to $\mathcal{P}\times\mathcal{F}$ such that for any $p\in\mathcal{P}$, $\mathbb{U}(p)$ is a configuration $(p',g)\in\mathcal{P}\times\mathcal{F}$, where

$$p'(\lambda) := \frac{1}{2}\left(p(\lambda)+p(1-\lambda)\right) \text{ and } g(\lambda) := \frac{p(\lambda)}{p(\lambda)+p(1-\lambda)} \tag{5}$$

**Lemma 4** *Let $(\mathbf{x},\mathbf{x}')_1^n$ be a sequence of sample pairs on which an Untied MixUp scheme specified by $(P^{\mathrm{uMix}},\gamma)$ and a DAT scheme with policy $P^{\mathrm{DAT}}$ will apply independently. If $(\mathbf{x},\mathbf{x}')_1^n$ is symmetric and $\left(P^{\mathrm{uMix}},\gamma\right)=\mathbb{U}(P^{\mathrm{DAT}})$, then $\mathcal{L}^{\mathrm{uMix}}\left((\mathbf{x},\mathbf{x}')_1^n,\gamma\right)=\mathcal{L}^{\mathrm{DAT}}\left((\mathbf{x},\mathbf{x}')_1^n\right)$.*

We now define another map $\mathbb{D}_u$ that maps an Untied MixUp scheme to a DAT scheme. Specifically $\mathbb{D}_u$ is a map from $\mathcal{P}\times\mathcal{F}$ to $\mathcal{P}$ such that for any $(p,g)\in\mathcal{P}\times\mathcal{F}$, $\mathbb{D}_u(p,g)$ is a configuration $p'\in\mathcal{P}$, where

$$p'(\lambda) := \left(g(\lambda)p(\lambda)+\overline{g(\overline{\lambda})}p(1-\lambda)\right)$$

It is easy to verify that $\int_0^1 p'(\lambda)d\lambda=1$. Thus $p'$ is indeed a distribution in $\mathcal{P}$ and $\mathbb{D}_u$ is well defined.

**Lemma 5** *Let $(\mathbf{x},\mathbf{x}')_1^n$ be a sequence of sample pairs on which an Untied MixUp scheme specified by $(P^{\mathrm{uMix}},\gamma)$ and a DAT scheme with policy $P^{\mathrm{DAT}}$ will apply independently. If $(\mathbf{x},\mathbf{x}')_1^n$ is symmetric and $P^{\mathrm{DAT}}=\mathbb{D}_u\left(P^{\mathrm{uMix}},\gamma\right)$, then $\mathcal{L}^{\mathrm{uMix}}\left((\mathbf{x},\mathbf{x}')_1^n,\gamma\right)=\mathcal{L}^{\mathrm{DAT}}\left((\mathbf{x},\mathbf{x}')_1^n\right)$.*

Lemmas 2, 3, 4 and 5 provide the building blocks for theorem 1, which we state hereafter. As $n$ increases, both DAT and UMixUp converge in probability toward their respective expected loss (lemmas 2 and 3). Since as $n$ increases, the sequence $(\mathbf{x},\mathbf{x}')_1^n$ becomes arbitrarily close to the symmetric sampling distribution $Q$, then by lemma 4 the family of DAT schemes converges in probability to a subset of UMixUp schemes. Lemma 5 proves the converse, i.e. that as $n$ increases the family of UMixUp schemes converges in probability to a subset of DAT schemes. **As $n$ increases, the family of UMixUp schemes therefore converges in probability to the entire family of DAT schemes**.

**Theorem 1** *Let $(X,X')_1^\infty$ be drawn i.i.d. from $Q$. On this sample-pair data, an Untied MixUp scheme specified by $(P^{\mathrm{Mix}},\gamma)$ and a DAT scheme specified by $P^{\mathrm{DAT}}$ will apply. In the Untied MixUp scheme, let $\Lambda_1^\infty$ be drawn i.i.d. from $P^{\mathrm{Mix}}$; in the DAT scheme, let $\Upsilon_1^\infty$ be drawn i.i.d. from $P^{\mathrm{DAT}}$. If $P^{\mathrm{DAT}}=\mathbb{D}_u\left(P^{\mathrm{Mix}},\gamma\right)$ or $\left(P^{\mathrm{Mix}},\gamma\right)=\mathbb{U}(P^{\mathrm{DAT}})$, then $\left|\mathcal{L}^{\mathrm{Mix}}\left((X,X')_1^n,\Lambda_1^n,\gamma\right)-\mathcal{L}^{\mathrm{DAT}}\left((X,X')_1^n,\Upsilon_1^n\right)\right|\xrightarrow{\mathrm{p}}0$, as $n\to\infty$*

The equivalence between the two families of schemes also indicates that there are DAT schemes that do not correspond to a MixUp scheme. These DAT schemes correspond to Untied MixUp scheme beyond the standard MixUp. The relationship between MixUp, DAT and Untied MixUp is shown in Figure 1.

## 6 UMixUp as a Useful Generalization of MixUp: Experiments

### 6.1 Experiment Setup and Implementation

We consider an image classification task on the Cifar10, Cifar100, MNIST and Fashion-MNIST datasets. The baseline classifier chosen is PreActResNet18 (see Liu (2017)), noting the same choice is made by the authors of MixupZhang et al. (2017b).

Both MixUp and Untied MixUp are considered in the experiments. The MixUp policies are chosen as Beta distribution $B(\alpha,\beta)$. The Untied MixUp policy is taken as $\mathbb{U}(B(\alpha,\beta))$.

Two target-linear loss functions are essayed: cross-entropy (CE) loss and the negative-cosine (CE) loss as defined earlier. We implement CE loss similarly to previous works, which use CE loss to implement the baseline model. In our implementation of the NC loss model, for each label $y$, $\varphi(y)$ is mapped to a randomly selected unit-length vector of dimension $d$ and fixed during training; the feature map of the original PreActResNet18 is linearly transformed to a $d$-dimensional vector. The dimension $d$ is chosen as 300 for Cifar10, MNIST and Fashion-Mnist (which have one black-and-white channel) and 700 for Cifar100 (which has 3 colored channels).

Our implementation of MixUp and Untied MixUp improves upon the published implementation from the original authors of MixUp Zhang et al. (2017b). For example, the original authors' implementation samples only one $\lambda$ per mini-batch, giving rise to unnecessarily higher stochasticity of the gradient signal. Our implementation samples $\lambda$ independently for each sample. Additionally, the original code combines inputs by mixing a mini-batch of samples with a shuffled version of itself. This approach introduces a dependency between sampled pairs and again increases the stochasticity of training. Our implementation creates two shuffled copies of the entire training dataset prior to each epoch, pairs them up, and then splits them into mini-batches. This gives a closer approximation to i.i.d. sampling and makes training smoother. While these implementation improvements have merit on their own, they do not provide a theoretical leap in understanding, and so we do not quantify their impact in our results analysis.

All models examined are trained using mini-batched backpropagation, for 200 epochs.

## 6.2 RESULTS

We sweep over the policy space of MixUp and Untied MixUp. For MixUp, it is sufficient to consider distribution $P^{\text{Mix}}$ to be symmetric about 0.5. Thus we consider only consider $P^{\text{Mix}}$ in the form of $\mathrm{B}(\alpha, \alpha)$, and scan through a single parameter $\alpha$ systematically. Since the policy of Untied MixUp is in the form of $\mathbb{U}(\mathrm{B}(\alpha, \beta))$, searching through $(\alpha, \beta)$ becomes more difficult. Thus our policy search for Untied MixUp is restricted to an *ad hoc* heuristic search. For this reason, the found best policy for Untied MixUp might be quite far from the true optimal.

The main results of our experiments are given in tables 1 to 4. As shown in the tables, each setting is run 100 times. For each run, we compute the error rate in a run as the average test error rate over the final 10 epochs. The estimated mean ("MEAN") performance of a setting is computed as the average of the error rates over all runs for the same setting. The 95%-confidence interval ("ConfInt") for the estimated mean performance is also computed and shown in the table.

From these results, we see that the Untied MixUp schemes each outperform their MixUp counterparts. Specifically, in 6 of the 8 cases (those printed in **bold font**), the confidence interval of Untied MixUp is completely disjoint from that of the corresponding MixUp scheme; and in some cases, the separation of confidence intervals is by a large margin. Note that the baseline model (PreActResNet18) has been designed with highly focused inductive bias for image classification tasks. Under such an inductive bias, one expects that the room for regularization (or the "amount of overfitting") isn't abundant. As such, we consider the improvement of Untied MixUp over MixUp rather significant.

The results show empirically that MixUp and Untied MixUp both work on the NC loss models. This validates our generalization of MixUp (and Untied MixUp) to models built with target linear losses.

| model | policy | runs | MEAN | ConfInt |
|---|---|---|---|---|
| baseline-CE | – | 100 | 5.476% | 0.027% |
| mixUp-CE | $\mathrm{B}(0.9, 0.9)$ | 100 | 4.199% | 0.023% |
| uMixUp-CE | $\mathbb{U}(\mathrm{B}(2.2, 0.9))$ | 100 | 4.177% | 0.025% |
| baseline-NC | – | 100 | 5.605% | 0.030% |
| mixUp-NC | $\mathrm{B}(1.0, 1.0)$ | 100 | 4.508% | 0.022% |
| **uMixUp-NC** | $\mathbb{U}(\mathrm{B}(1.8, 1.0))$ | 100 | **4.455%** | **0.025%** |

Table 1: Test error rate on CIFAR10.

| model | policy | runs | MEAN | ConfInt |
|---|---|---|---|---|
| baseline-CE | − | 100 | 24.848% | 0.060% |
| mixUp-CE | $B(0.9, 0.9)$ | 100 | 22.020% | 0.050% |
| **uMixUp-CE** | $\mathbb{U}(B(1.4, 0.7))$ | 100 | **21.884%** | **0.051%** |
| baseline-NC | − | 100 | 25.270% | 0.050% |
| mixUp-NC | $B(0.9, 0.9)$ | 100 | 24.298% | 0.051% |
| **uMixUp-NC** | $\mathbb{U}(B(1.3, 0.9))$ | 100 | **23.819%** | **0.054%** |

Table 2: Test error rate on CIFAR100.

| model | policy | runs | MEAN | ConfInt |
|---|---|---|---|---|
| baseline-CE | − | 100 | 0.816% | 0.007% |
| mixUp-CE | $B(1.0, 1.0)$ | 100 | 0.632% | 0.005% |
| **uMixUp-CE** | $\mathbb{U}(B(1.7, 1.0))$ | 100 | **0.609%** | **0.005%** |
| baseline-NC | − | 100 | 0.720% | 0.007% |
| mixUp-NC | $B(1.0, 1.0)$ | 100 | 0.607% | 0.004% |
| **uMixUp-NC** | $\mathbb{U}(B(1.3, 0.9))$ | 100 | **0.592%** | **0.005%** |

Table 3: Test error rate on MNIST.

| model | policy | runs | MEAN | ConfInt |
|---|---|---|---|---|
| baseline-CE | − | 100 | 5.060% | 0.027% |
| mixUp-CE | $B(1.0, 1.0)$ | 100 | 4.585% | 0.013% |
| uMixUp-CE | $\mathbb{U}(B(1.7, 0.8))$ | 100 | 4.570% | 0.013% |
| baseline-NC | − | 100 | 5.083% | 0.016% |
| mixUp-NC | $B(1.0, 1.0)$ | 100 | 4.767% | 0.013% |
| **uMixUp-NC** | $\mathbb{U}(B(1.3, 0.9))$ | 100 | **4.613%** | **0.011%** |

Table 4: Test error rate on Fashion-MNIST.

## 7  CONCLUDING REMARKS

This paper establishes a connection between MixUp and adversarial training. This connection allows for a better understanding of the working mechanism of MixUp as well as a generalization of MixUp to a wider family, namely Untied MixUp. Despite the development in this work, it is the authors' belief that the current designs of MixUp and Untied MixUp are far from optimal. In particular, we believe a better design should allow individualized policy for each training pair. How this can be done remains open at this time.

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

# A APPENDIX

## A.1 PROOF OF LEMMA 2:

For any fixed infinite sequence $(\mathbf{x}, \mathbf{x}')_1^\infty$ of samples drawn i.i.d. from $Q$ and any infinite sequence of i.i.d. random variables $\Lambda_1^\infty$ drawn from $P^{\mathrm{DAT}}$, let $\mathcal{L}^{\mathrm{DAT}}((\mathbf{x}, \mathbf{x}')_1^n, \Lambda_1^n)$ be defined according to (4), with the first $n$ elements of $(\mathbf{x}, \mathbf{x}')_1^\infty$ and the first $n$ elements of $\Lambda_1^\infty$ as input. Define

$$
\delta_{\mathrm{DAT}} := \max_{\substack{(\mathbf{x},\mathbf{x}') \in \\ \mathcal{D} \times \mathcal{D}}} \sup_{\substack{(\lambda,\lambda') \in \\ [0,1] \times [0,1]}} |\ell^{\mathrm{DAT}}(\mathbf{x}, \mathbf{x}', \lambda) - \ell^{\mathrm{DAT}}(\mathbf{x}, \mathbf{x}', \lambda')|
$$

For any given $\lambda_1^n \in [0,1]^n$ and any of its modified version $u_1^n \in [0,1]^n$ which differs from $\lambda_1^n$ in exactly one location, it can be verified, following the definition of $\delta_{\mathrm{DAT}}$, that

$$
\left| \mathcal{L}^{\mathrm{DAT}}((\mathbf{x}, \mathbf{x}')_1^n, \lambda_1^n) - \mathcal{L}^{\mathrm{DAT}}((\mathbf{x}, \mathbf{x}')_1^n, u_1^n) \right| \le \delta_{\mathrm{DAT}}/n
$$

Since $\Lambda_1, \Lambda_2, \ldots \Lambda_K$ are independent and by McDiarmid Inequality McDiarmid (1989), it follows that for any $\epsilon > 0$,

$$
\Pr\left[ \mathcal{L}^{\mathrm{DAT}}((\mathbf{x}, \mathbf{x}')_1^n, \Lambda_1^n) - \mathcal{L}_E^{\mathrm{DAT}}((\mathbf{x}, \mathbf{x}')_1^n) \ge \epsilon \right] < 2\exp\left( -\frac{2\epsilon^2}{n \cdot (\delta_{\mathrm{DAT}}/n)^2} \right)
$$

which proves the lemma $\qquad\square$

## A.2 PROOF OF LEMMA 4:

$$
\begin{aligned}
\mathcal{L}_E^{\mathrm{uMix}}((\mathbf{x}, \mathbf{x}')_1^n, \gamma) &:= \frac{1}{n}\sum_{k=1}^n \mathbb{E}_{\lambda \sim P^{\mathrm{Mix}}}\left\{ \gamma(\lambda)\ell^{\mathrm{DAT}}(\mathbf{x}_k, \mathbf{x}'_k, \lambda) + \overline{\gamma(\lambda)}\ell^{\mathrm{DAT}}(\mathbf{x}'_k, \mathbf{x}_k, \overline{\lambda}) \right\} \\
&= \frac{1}{n}\sum_{k=1}^n \int \left( \gamma(\lambda)P^{\mathrm{Mix}}(\lambda)\ell^{\mathrm{DAT}}(\mathbf{x}_k, \mathbf{x}'_k, \lambda) + \overline{\gamma(\lambda)}P^{\mathrm{Mix}}(\lambda)\ell^{\mathrm{DAT}}(\mathbf{x}'_k, \mathbf{x}_k, \overline{\lambda}) \right) d\lambda \\
&= \frac{1}{n}\sum_{k=1}^n \int \left( \frac{1}{2}P^{\mathrm{DAT}}(\lambda)\ell^{\mathrm{DAT}}(\mathbf{x}_k, \mathbf{x}'_k, \lambda) + \frac{1}{2}P^{\mathrm{DAT}}(\overline{\lambda})\ell^{\mathrm{DAT}}(\mathbf{x}'_k, \mathbf{x}_k, \overline{\lambda}) \right) d\lambda \\
&= \frac{1}{2K}\left( \sum_{k=1}^n \int P^{\mathrm{DAT}}(\lambda)\ell^{\mathrm{DAT}}(\mathbf{x}_k, \mathbf{x}'_k, \lambda)d\lambda + \sum_{k=1}^n \int P^{\mathrm{DAT}}(\overline{\lambda})\ell^{\mathrm{DAT}}(\mathbf{x}'_k, \mathbf{x}_k, \overline{\lambda})d\lambda \right) \\
&\overset{(a)}{=} \frac{1}{2K}\left( \sum_{k=1}^n \int P^{\mathrm{DAT}}(\lambda)\ell^{\mathrm{DAT}}(\mathbf{x}_k, \mathbf{x}'_k, \lambda)d\lambda + \sum_{k=1}^n \int P^{\mathrm{DAT}}(\lambda)\ell^{\mathrm{DAT}}(\mathbf{x}'_k, \mathbf{x}_k, \lambda)d\lambda \right) \\
&\overset{(b)}{=} \frac{1}{2K}\left( \sum_{k=1}^n \int P^{\mathrm{DAT}}(\lambda)\ell^{\mathrm{DAT}}(\mathbf{x}_k, \mathbf{x}'_k, \lambda)d\lambda + \sum_{k=1}^n \int P^{\mathrm{DAT}}(\lambda)\ell^{\mathrm{DAT}}(\mathbf{x}_k, \mathbf{x}'_k, \lambda)d\lambda \right) \\
&= \frac{1}{n}\sum_{k=1}^n \int P^{\mathrm{DAT}}(\lambda)\ell^{\mathrm{DAT}}(\mathbf{x}_k, \mathbf{x}'_k, \lambda)d\lambda \\
&= \mathcal{L}_E^{\mathrm{DAT}}((\mathbf{x}, \mathbf{x}')_1^n)
\end{aligned}
$$

where (a) is due to a change of variable in the integration, (b) is due to the symmetry of $(\mathbf{x}, \mathbf{x}')_1^n$. Note that in equation 5 $g(\lambda)$ is undefined at values of $\lambda$ for which the denominator is zero. But the lemma holds true because the denominator is only zero when $p(\lambda) = 0$, so those $\lambda$ for which $g(\lambda)$ is undefined never get drawn in the DAT scheme. $\qquad\square$

### A.3 PROOF OF LEMMA 5:

$$
\begin{aligned}
\mathcal{L}_E^{\text{uMix}}\left((\mathbf{x},\mathbf{x}')_1^n,\gamma\right) &= \frac{1}{n}\mathbb{E}_{\lambda\sim P^{\text{Mix}}}\sum_{k=1}^n\left(\gamma(\lambda)\ell^{\text{DAT}}(\mathbf{x}_k,\mathbf{x}'_k,\lambda)+\overline{\gamma(\lambda)}\ell^{\text{DAT}}(\mathbf{x}'_k,\mathbf{x}_k,\overline{\lambda})\right) \\
&= \frac{1}{n}\left(\mathbb{E}_{\lambda\sim P^{\text{Mix}}}\sum_{k=1}^n\gamma(\lambda)\ell^{\text{DAT}}(\mathbf{x}_k,\mathbf{x}'_k,\lambda)+\mathbb{E}_{\lambda\sim P^{\text{Mix}}}\sum_{k=1}^n\overline{\gamma(\lambda)}\ell^{\text{DAT}}(\mathbf{x}'_k,\mathbf{x}_k,\overline{\lambda})\right) \\
&\overset{(a)}{=} \frac{1}{n}\left(\mathbb{E}_{\lambda\sim P^{\text{Mix}}}\sum_{k=1}^n\gamma(\lambda)\ell^{\text{DAT}}(\mathbf{x}_k,\mathbf{x}'_k,\lambda)+\mathbb{E}_{\lambda\sim P^{\text{Mix}}}\sum_{k=1}^n\overline{\gamma(\lambda)}\ell^{\text{DAT}}(\mathbf{x}_k,\mathbf{x}'_k,\overline{\lambda})\right) \\
&\overset{(b)}{=} \frac{1}{n}\left(\mathbb{E}_{\lambda\sim P^{\text{Mix}}}\sum_{k=1}^n\gamma(\lambda)\ell^{\text{DAT}}(\mathbf{x}_k,\mathbf{x}'_k,\lambda)+\mathbb{E}_{\overline{\lambda}\sim P^{\text{Mix}}}\sum_{k=1}^n\overline{\gamma(\overline{\lambda})}\ell^{\text{DAT}}(\mathbf{x}_k,\mathbf{x}'_k,\lambda)\right) \\
&= \frac{1}{n}\sum_{k=1}^n\int\left(\gamma(\lambda)P^{\text{Mix}}(\lambda)\ell^{\text{DAT}}(\mathbf{x}_k,\mathbf{x}'_k,\lambda)+\overline{\gamma(\overline{\lambda})}P^{\text{Mix}}(1-\lambda)\ell^{\text{DAT}}(\mathbf{x}_k,\mathbf{x}'_k,\lambda)\right)d\lambda \\
&= \frac{1}{n}\sum_{k=1}^n\int\ell^{\text{DAT}}(\mathbf{x}_k,\mathbf{x}'_k,\lambda)\underbrace{\left(\gamma(\lambda)P^{\text{Mix}}(\lambda)+\overline{\gamma(\overline{\lambda})}P^{\text{Mix}}(1-\lambda)\right)}_{\mathbb{D}_{\text{u}}(P^{\text{Mix}},\gamma)}d\lambda \\
&= \frac{1}{n}\sum_{k=1}^n\mathbb{E}_{\lambda\sim P^{\text{DAT}}}\ell^{\text{DAT}}(\mathbf{x}_k,\mathbf{x}'_k,\lambda) \\
&= \mathcal{L}_E^{\text{DAT}}\left((\mathbf{x},\mathbf{x}')_1^n\right).
\end{aligned}
$$

where (a) is due to the symmetry of $(\mathbf{x},\mathbf{x}')_1^n$, and (b) is by a change of variable in the second term (renaming $1-\lambda$ as $\lambda$). $\qquad\square$

