# OpenReview forum: "MixUp as Directional Adversarial Training"
_ICLR.cc/2020/Conference — Reject_

### Official Review · AnonReviewer2 · 2019-10-19
**Official Blind Review #2**

**Rating:** 1

**Review:**

First of all, the concept of 'Directional Adversarial Training (DAT)' is not appropriate. Actually similar method has been proposed in Hiroshi Inoue (2018) as a data augmentation method.

What surprised me the most is that after trying to connect UMixUp with adversarial training in the whole paper, there is no evaluation of adversarial robustness in the experiments? The main character of adversarial training is an improvement in robustness and degeneration on clean accuracy, which is different from the performance of UMixup or Mixup.

The authors should do a major modification on their motivation and experiments before the paper is able to be published.

Reference:
[1] Hiroshi Inoue. Data Augmentation by Pairing Samples for Images Classification. arXiv 1801.02929


**Experience Assessment:**

I have published in this field for several years.

**Review Assessment: Checking Correctness Of Derivations And Theory:**

I assessed the sensibility of the derivations and theory.

**Review Assessment: Checking Correctness Of Experiments:**

I carefully checked the experiments.

**Review Assessment: Thoroughness In Paper Reading:**

I made a quick assessment of this paper.

---

> ### Author Response · Authors · 2019-11-10
> **Made a quick assessment of this paper, eh?**
>
> "First of all, the concept of 'Directional Adversarial Training (DAT)' is not appropriate. Actually similar method has been proposed in Hiroshi Inoue (2018) as a data augmentation method."
>
> 1) Why is it not appropriate?  If there is a similar scheme to ours, then the concept of DAT would not be appropriate? We do not see your rationale.
> 2) We agree that  Inoue's scheme falls into the general scope related to this work , and should be cited in our work.
> 3) We agree Inoue's bears some similarity, but it is different from DAT. For example, Inoue's scheme mixes input samples with a mixing ratio of 0.5, while we consider all convex combinations; Inoue's scheme mixes input samples after performing other regularizations, while we mix ours directly.
>
> "What surprised me the most is that after trying to connect UMixUp with adversarial training in the whole paper, there is no evaluation of adversarial robustness in the experiments? The main character of adversarial training is an improvement in robustness and degeneration on clean accuracy, which is different from the performance of UMixup or Mixup."
>
> There are two related research themes in this context. One is on schemes to improve adversarial robustness, the other is on schemes to improve generalization performance. The interest of the paper is the latter, not the former. The fact that we are able to connect MixUp to adversarial training does not mean we are at all interested in adversarial robustness.

---

> > ### Comment · AnonReviewer2 · 2019-11-10
> > **Yes, a quick assessment is enough for this paper.**
> >
> > I assume that the authors know what is adversarial training. On CIFAR-10, adversarial training will decrease the clean accuracy from ~94% to ~83%, i.e., adversarial training will largely degenerate the generalization performance. As compensation, adversarial training leads to better robustness.
> >
> > Can the authors explain why UMixup is analogous to adversarial training, but have totally contradictory properties?
> >
> > Besides, if you do not focus on improving robustness, then what you want to demonstrate by relating UMixup with adversarial training?

---

> > > ### Author Response · Authors · 2019-11-10
> > > **a quick reply to your comments**
> > >
> > > "adversarial training will largely degenerate the generalization performance".
> > >
> > > We do not think this statement is true.  In general adversarial robustness and generalization capability appear to be correlated concepts.  When the model overfits, adversarial training schemes (particularly those based on min-max optimization of the loss function) can be used as regularization schemes,  often superior to weight decay or dropout. We have observed this in many applications.  Similar results are also reported in the literature, e.g., in Goodfellow et al "Explaining and Harnessing Adversarial Examples".
> > >
> > > "Can the authors explain why UMixup is analogous to adversarial training, but have totally contradictory properties?"
> > >
> > > We did explain why UMixUp and MixUp are similar to adversarial training in the paper, with lengthy proofs. They might require a bit more than "a quick assessment" to understand.  Regarding your statement that UMixUp (or MixUp) and adversarial training "have totally contradictory properties", we do not think it is well-supported.
> > >
> > > "Besides, if you do not focus on improving robustness, then what you want to demonstrate by relating UMixup with adversarial training?"
> > >
> > > We would like to show that MixUp, as a regularization scheme, takes effect in a way similar to how adversarial training regularizes models.

---

> > > ### Author Response · Authors · 2019-11-10
> > > **Addressing the reviewer's concerns**
> > >
> > > Hello, I am another co-author of this paper.
> > >
> > > "On CIFAR-10, adversarial training will decrease the clean accuracy from ~94% to ~83%, i.e., adversarial training will largely degenerate the generalization performance. As compensation, adversarial training leads to better robustness. "
> > >
> > > The reviewer is mistaken in assuming that adversarial training schemes necessarily achieve adversarial robustness at the cost of generalization, or that there is a tradeoff between robustness and accuracy.
> > >
> > > While there is not full consensus on the functional mechanisms of adversaral training, adversarial training schemes are generally agreed to reduce overfitting. Schemes that seek not merely to improve, but to maximize adversarial robustness will then become underfit, since they reduce overfitting too aggressively. This results in degraded accuracy.
> > >
> > > However, properly tuned adversarial schemes will reduce overfitting to just the degree required for a good fit. As cited by my co-author, Goodfellow's experiments showed that adversarial training can even outperform dropout in terms of classification accuracy.
> > >
> > > "Can the authors explain why UMixup is analogous to adversarial training, but have totally contradictory properties?"
> > >
> > > If the reviewer's primary concern is that we have not shown that UMixUp improves adversarial robustness, they are mistaken:
> > > 1) MixUp improves adversarial robustness, as shown by Lamb et al. and cited in our paper
> > > 2) UMixUp, as a superset of MixUp, is therefore guaranteed to improve adversarial robustness for at least one member (namely MixUp).
> > >
> > >
> > > If, finally, the reviewer's concern is that we have not shown that hat at least one member of UMixUp improves adversarial robustness over MixUp, we are not opposed to this as a future addition. However, given the self-evident conclusion that a superset of MixUp will improve over MixUp, and given that our research interest is in improving generalization rather than robustness, we believe this would only marginally contribute to our paper.

---

> > > > ### Comment · AnonReviewer2 · 2019-11-10
> > > > **Thank you for the placid response**
> > > >
> > > > According to the authors' rebuttal,  I suppose that the authors are not quite familiar with the research topic on the adversarial robustness. Below I will support my points by some widely accepted papers.
> > > >
> > > > My conclusion of "On CIFAR-10, adversarial training will decrease the clean accuracy from ~94% to ~83%" is based on [1] (Figure 4), which is the most popular adversarial training scheme now. (Almost 1,000 citations on Google Scholar)
> > > >
> > > > The paper of Goodfellow et al. is indeed one of the pioneers and seminal work on the problem of adversarial examples. However, it has been five years after its publication, and many later papers show that some of the conclusions in Goodfellow et al. (2015) are not necessarily true. For example, the trade-off between clean accuracy and adversarial robustness is verified both theoretically and empirically [2][3].
> > > >
> > > > Besides, if the authors really read the work of Lamb et al. before citing it, you should directly see the trade-off by its title:" Interpolated Adversarial Training: Achieving Robust Neural Networks without Sacrificing Too Much Accuracy". The motivation of them to combine mixup with adversarial training is exactly because adversarial training will sacrifice clean accuracy. And the role of the mixup in interpolated adversarial training is not to further improve robustness, but improve clean accuracy.
> > > >
> > > >
> > > >
> > > > References:
> > > > [1] Madry et al. Towards Deep Learning Models Resistant to Adversarial Attacks. ICLR 2018
> > > > [2] Tsipras et al. Robustness May Be at Odds with Accuracy. ICLR 2019
> > > > [3] Zhang et al. Theoretically Principled Trade-off between Robustness and Accuracy. ICML 2019

---

> > > > > ### Author Response · Authors · 2019-11-10
> > > > > **Reply**
> > > > >
> > > > > Although the main interest of this work is not adversarial robustness, we are in fact aware of the papers you brought up. As a short overall comment, we believe that with the current understanding of adversarial robustness and generalization in the deep learning community, it is too early to conclude that there is unconditional tradeoff between improving generalization and improving  adversarial robustness. (Here “unconditional” means that such a tradeoff is independent of all or some of the following: the definition of adversarial robustness, the distribution of the data, the sample size, the model capacity, and the training method.) On the contrary, there does appear a correlation between improving generalization and improving adversarial robustness.
> > > > >
> > > > > In Tsipras et al. “Robustness May Be at Odds with Accuracy”, it is clearly shown that when the number of training examples is small, adversarial training helps generalization.  According to the paper, the tension between accuracy and adversarial robustness only arises after the number of training examples exceed some level. Note that when the number of training examples is small, this is also the regime where the model overfits and regularization is effective.
> > > > >
> > > > > In Zhang et al. “Theoretically Principled Trade-off between Robustness and Accuracy”, a notion of adversarial robustness is defined and, under some condition, a theoretical tradeoff between such robustness and the classification accuracy is proved. We wish to point out however that the notion of adversarial robustness there isn't necessarily the target to which the popular adversarial training schemes (such as Goodfellow's FGSM or Madry et al's PGD) try to drive the model. In fact, for a consideration like such, the authors propose a different adversarial training scheme. Additionally, we note that in their development, the potential difference between the model's performance on the training data and its performance on the testing data, namely, the model’s generalization behaviour, is not touched upon. Thus, the proved tradeoff between robustness and accuracy should not be understood as a tradeoff between robustness and generalization.
> > > > >
> > > > > Overall, we are still far from understanding the behaviour of PGD-like adversarial training and understanding the precise relationship between adversarial robustness and generalization. Nonetheless, in the sparse sample regime, one may sketch an intuitive picture: the lack of training examples makes the learned decision boundary too close to the training examples. This phenomenon causes the model to overfit, and this same phenomenon also causes the model to be fragile against adversarial attacks. Thus, at least in this regime, regularization and adversarial defence appear to have highly correlated objectives, namely, pushing the decision boundary away from the training examples. When the number of samples increases, the two objectives may diverge, potentially causing a tension between the two.

---

### Official Review · AnonReviewer3 · 2019-10-23
**Official Blind Review #3**

**Rating:** 3

**Review:**

This paper proposes a novel data augmentation method, untied MixUp (UMixUp), which is a general case of both MixUp and Directional Adversarial Traning (DAT). DAT is referred to in this paper as a scheme that only input feature vectors are mixed, while MixUp also incorporates their corresponding labels. The authors provide a theoretical discussion that both DAT and UMixUp converges to be equivalent to each other when the number of training samples becomes infinity. Experimental results on Cifar 10, Cifar 100, MNIST, and Fashion MNIST show quantitative comparisons among the baseline, MixUp, and UMixUp.

According to the author guideline,
> There will be a strict upper limit of 10 pages for the main text. Reviewers will be instructed to apply a higher standard to papers in excess of 8 pages.
The authors use nine pages. Therefore, the review should be more careful about its quality.

Currently, I have three major concerns that keep me from judging this paper acceptable in ICLR 2020.

First, the authors failed to cite two closely related papers below:
- Tokozume et al., LEARNING FROM BETWEEN-CLASS EXAMPLES FOR DEEP SOUND RECOGNITION. ICLR, 2018.
- Tokozume et al., Between-class Learning for Image Classification. CVPR, 2018.
The first one is published in the previous ICLR and mixing two samples belonging to different classes. The second one is an application to image classification using ImageNet dataset, which is larger than the dataset used in this paper.  What's more important is that both papers propose that the mixing ratio of two samples is not linearly but depending on the strength of their signals. Since UMixUp is also focusing on the mixing ratio between two training samples, Between-Class Learning should have been compared to the proposed method.

Secondly, the theoretical discussion is not so fascinating. Actually, both MixUp and UMixUp are shown to converge to DAT when the number of training samples tends to infinity. Data augmentation is, however, performed to remedy the lack of training samples in general. The discussion that the number of training samples is assumed to be large is the opposite situation.

Thirdly, the experimental results show that the performance gain by UMixUp is relatively small in comparison to that of the original MixUp. There are no ablation studies using different values for alpha and beta, which are parameters for the policy of UMixUp. The authors reported that these values are defined using a heuristic search. Thus, we cannot see if the performance is sensitive to the parameter selection.

I lean to reject this paper because of these concerns. I'm looking forward to seeing the revised version in another conference.

**Experience Assessment:**

I have published in this field for several years.

**Review Assessment: Checking Correctness Of Derivations And Theory:**

I assessed the sensibility of the derivations and theory.

**Review Assessment: Checking Correctness Of Experiments:**

I carefully checked the experiments.

**Review Assessment: Thoroughness In Paper Reading:**

I read the paper thoroughly.

---

> ### Author Response · Authors · 2019-11-10
> **Thanks for your review, but for most, we do not agree, either.**
>
> " Between-Class Learning should have been compared to the proposed method"
>
> We agree that BC Learning merits a citation. However we do not share the reviewer's view that it is of high relevance to our paper and must be compared against.   In particular, we note that BC learning of Tukozume et al. performs mixing in a different manner than MixUp and Untied MixUp. The difference is summarized below.
>
> 1) While label mixing is a convex combination, like MixUp, BC learning’s input mixing is non a convex combination. This has the  critical geometric interpretation that the family of input mixes follows a curve between the mixed samples rather than a straight line.
> 2) MixUp proposes a family of distributions for ratio of sampling, whereas BC learning proposes only the uniform distribution.
> 3) BC Learning proposes a specific label mixing function back by physical properties (of sounds and images), rather than a general framework for mixing samples for datasets in any field
> 4) Finally, BC learning’s introduction of a non-linear function in mixing is markedly different than Untied MixUp. BC learning uses a specific non-linear function of the label mixing ratio to arrive at the input mixing ratio. By contrast, Untied MixUp uses a family of non-linear functions of the input mixing ratio to arrive at the label mixing ratio.
>
> In general, we feel that the similarities between BC Learning and MixUp, and especially Untied MixUp, are limited to the fact that both works perform input and label mixing. This work mainly aims at explaining MixUp, not BC learning, we do not think the added value from comparing with BC learning is essential to justify the acceptance of this paper.
>
> "the theoretical discussion is not so fascinating"
>
> Fascinating or not is a matter of subjective taste, and we subjectively disagree with this comment.
>
> "Actually, both MixUp and UMixUp are shown to converge to DAT when the number of training samples tends to infinity. Data augmentation is, however, performed to remedy the lack of training samples in general. The discussion that the number of training samples is assumed to be large is the opposite situation."
>
> The reviewer is mistaken here. First, the quantity n in the paper is not the number of training examples, but the number of example pairs drawn the training set. This is stated in the second paragraph of Section 3.2, when the quantity n appears for the first time in the paper. As a matter of fact, nowhere in the paper, we have any specific requirement on the size of the training set.  That is, all theoretical results in the paper are proved for training set of arbitrary sizes.
>
> "the experimental results show that the performance gain by UMixUp is relatively small in comparison to that of the original MixUp."
>
> One can run into endless debate on what kind of improvement is considered "not small", particularly in light that the  baseline scheme and the compared MixUp scheme are already very strong, leaving little room for further improvement. Instead of debating on this, we would like to re-iterate that the improvements of Untied MixUp over MixUp  are significant in 6 out of the 8 examined cases, where the significance is demonstrated by the disjoint confidence intervals of the two schemes (more quantitative demonstration of such significance is also possible).  Nonetheless, we wish to re-emphasize that the proposal of Untied MixUp and the performance gain it may bring are only a minor contribution of this paper. Our main contribution lies in connecting MixUp with a special form of adversarial training (DAT) and showing the existence of a superset (Untied MixUp) that possesses the same adversarial intuition as MixUp.
>
> "There are no ablation studies using different values for alpha and beta, which are parameters for the policy of UMixUp. The authors reported that these values are defined using a heuristic search. Thus, we cannot see if the performance is sensitive to the parameter selection."
>
> We agree that our experimentation procedure is inadequately described. In fact, our method was revamped since the our submission of this paper to an earlier publication, and we omitted to update our methodology for the present ICLR submission. We do pride ourselves on the quality of our experiments, and the rigour of our methods. We will describe our experimentation procedure in greater detail in our revised paper if published in ICLR.

---

### Official Review · AnonReviewer1 · 2019-10-26
**Official Blind Review #1**

**Rating:** 3

**Review:**

This paper introduces directional adversarial training (DAT) and UMixUP, which are extension methods of MixUp. DAT and UMixUp use the same method of MixUp for generating samples but use different label mixing ratios where DAT retains the sample's original label. In contrast, UMixUp uses a function of the input mixing ratio. This paper shows that UMixUp and DAT are equivalent when the number of samples tends to infinity. In the experiments, UMixUp provides an improvement over MixUp.

This paper should be rejected because the originality of the proposed method over MixUp is marginal, and the improvement of classification accuracy is not surprising, although the explanation of the relationship among DAT, UMixUp, and the original MixUp is nice. The modification of label mixing ratios is not enough contribution. A more precise description of why DAT and UMixUp work better over the original MixUp is required.

The same idea of MixUp was proposed at the same conference (ICLR2018). It should be cited.
Tokozume et al., Learning from Between-class Examples for Deep Sound Recognition. ICLR, 2018.
Tokozume et al. explain why the mixture of examples works well from the different perspectives of adversarial examples. Also, BC learning uses a KL loss, not a cross-entropy loss. Therefore, the reviewer doubts the following statement: "MixUp is only applicable to baseline models that use cross entropy loss" on page 2.

Minor comments
1) In page 8, "negative-cosine (CE)" might be "negative-cosine　(NC)".

**Experience Assessment:**

I have published in this field for several years.

**Review Assessment: Checking Correctness Of Derivations And Theory:**

I assessed the sensibility of the derivations and theory.

**Review Assessment: Checking Correctness Of Experiments:**

I assessed the sensibility of the experiments.

**Review Assessment: Thoroughness In Paper Reading:**

I read the paper at least twice and used my best judgement in assessing the paper.

---

> ### Author Response · Authors · 2019-11-10
> **Thanks for your review, but for most of it,  we do not agree**
>
> “the originality of the proposed method over MixUp is marginal"
>
> Stripped of its association with adversarial training, Untied MixUp might not be sufficiently interesting. But the real contribution of our paper is not about introducing Untied MixUp, but explaining that adversarial training is a fundamental working mechanism of MixUp and that a larger set of schemes benefit from the same explanatory power. It is only given the key insight that Untied MixUp corresponds precisely to this superset of adversarial-like schemes that its simplicity and accessible intuition become an elegant upshot, rather than a prosaic observation unworthy of publication. Even though advocating Untied MixUp is not the main purpose of this paper, we still consider its improvement over MixUp significant. In 6 out of the 8 compared cases, Untied MixUp outperform MixUp with disjoint confidence intervals.
>
> "the improvement of classification accuracy is not surprising”
>
> Indeed the improvement of Untied MixUp over MixUp is expected. It results from our analysis and discovered connection between MixUp and DAT. Showing the improvement serves to further validate our analysis and discovery.
>
> “The modification of label mixing ratios is not enough contribution".
>
> If the paper only contained the proposal of Untied Mixup, we would agree that it does not have enough contribution. But the key message of the paper is the discovered connection between MixUp and adversarial training. The proposal of Untied MixUp is just a by-product.
>
> "A more precise description of why DAT and UMixUp work better over the original MixUp is required.”
>
> To be clear, DAT’s performance accuracy is *not* superior to MixUp. In the paper, we never said DAT is better. This is because, as we show in the paper, only when summed over an infinitely large batch (aka n goes to infinity) of DAT-perturbed examples will the DAT loss function converges to the Untied MixUp loss. However for small batch sizes (n small) used in practice, the loss function of DAT will be more volatile, which in fact gives slower training and worse results than MixUp.
>
> As for the review's request for more a precise description as to why Untied MixUp is works better than the original MixUp, we do not believe this is necessary. As the reviewer herself (or himself) noted and we addressed above, the improvement is "not surprising", or, obvious. Indeed, it is: Untied MixUp is a superset of MixUp, necessarily containing a better member.
>
> “The same idea of MixUp was proposed at the same conference (ICLR2018). It should be cited."
>
> Indeed BC learning lies in the general scope of relevance. We will cite it in the revision. But we do not think the existence of BC learning puts any negative shadow on this paper.
>
> " Also, BC learning uses a KL loss, not a cross-entropy loss. Therefore, the reviewer doubts the following statement: "MixUp is only applicable to baseline models that use cross entropy loss" on page 2."
>
>
> Your quote is missing the first word of our sentence. Our full sentence was "Conventionally, MixUp is only applicable to baseline models that use cross entropy loss".  In this sentence, MixUp refers to the work of (Zhang et al 2017), which introduces a method they call  "MixUp". That is, we speak of MixUp in its original sense.

---

### Public Comment · ~Bao_Wang1 · 2019-10-19
**An interesting work**

Hi,
   It is an interesting idea to consider interpolation in adversarial training. I would like to point out a few closely related papers to you, and I wish they are of interest to you.

1. B. Wang, et al. Deep Neural Nets with Interpolating Function as Output Activation, NeurIPS 2018.

2. B. Wang, et al. Adversarial Defense via Data Dependent Activation Function and Total Variation Minimization, arXiv:1809.08516 2018

3. B. Wang, et al. Graph Interpolating Activation Improves Both Natural and Robust Accuracies in Data-Efficient Deep Learning, arXiv:1907.06800 2019.

Thanks for your attention.

---

### Decision · Program_Chairs · 2019-12-19

**Decision:**

Reject

**Comment:**

This paper builds a connection between MixUp and adversarial training. It introduces untied MixUp (UMixUp), which generalizes the methods of MixUp. Then, it also shows that DAT and UMixUp use the same method of MixUp for generating samples but use different label mixing ratios. Though it has some valuable theoretical contributions, I agree with the reviewers that it’s important to include results on adversarial robustness, where both adversarial training and MixUp are playing an important role.